# Influence of Adding Offcuts and Trims with a Recycling Approach on the Properties of High-Density Fibrous Composites

**DOI:** 10.3390/polym12061327

**Published:** 2020-06-10

**Authors:** Conrad M. Sala, Eduardo Robles, Grzegorz Kowaluk

**Affiliations:** 1Department of Technology and Entrepreneurship in Wood Industry, Warsaw University of Life Sciences—SGGW, Nowoursynowska St. 159, 02-776 Warsaw, Poland; conrad_sala@sggw.edu.pl; 2Chief Technologist, IKEA Industry Poland sp. z o. o. brand Orla, Koszki 90, 17-106 Orla, Poland; 3Institute of Analytical and Physicochemical Sciences for the Environment and Materials (IPREM-UMR 5254), CNRS/Univ Pau & Pays Adour/E2S UPPA, IUT of the Pays de l’Adour, 371 Rue du Ruisseau, 40004 Mont de Marsan, France; eduardo.robles@univ-pau.fr

**Keywords:** fibers, recycling, wood, mechanical properties, physical properties, strength

## Abstract

The sizeable global production of wood-based products requires new sources of raw material, but also creates large quantities of wastes or composites that do not comply with requirements. In this study, the influence of different shares of recovered high-density fiberboards (HDF-r), reversed into the production, on industrial HDF properties, has been examined. HDF-r may be a suitable partial substitute for raw pinewood for industrial HDF production. Although most of the mechanical properties, as well as thickness swelling and water absorption, had a linear decrease with the increase in the share of HDF-r share, the elaborated boards met most of the commercial requirements (EN 622-5). The property that did not meet the requirements was the internal bond strength for panels with 10% of HDF-r. The presented results show that, after some adjustments, it should be possible to produce HDF boards with up to 10% of recycled HDF being able to meet all commercial requirements.

## 1. Introduction

Global development and population growth [1] result in a more significant demand for new space for accommodation. Due to that fact, there is also a growing need for building materials, not only bricks or plasterboards, but also wood-based panels, for example, oriented strand boards (OSB), plywood, or insulation boards, including soft boards (SB) or low-density fiberboards (LDF). Together with the operation new residential buildings, the demand for new furniture made of, for example, particleboards (PB) or fiberboards, among which the most common are hardboard (HB), medium-density fiberboards (MDF), and high-density fiberboards (HDF), is growing as well. For this reason, there has been a considerable growth in the world production of wood-based panels. Figure 1 presents how the world production of wood-based panels has increased from 124 million m^3^ in 1990 to over 405 million m^3^ 2018 [2].

The primary material for the production of wood-based panels is wood itself. Based on the data shown in Figure 1, it can be seen that the world production of coniferous pulpwood, round wood, and split wood has grown from 266 million m^3^ in 2000 to over 351 million m^3^ in 2018 [2]. The price increase of the raw materials has supposed a struggle for the furniture industry [3]. In Poland, in 2018, a significant jump (39.2%) in the price of building materials was observed, in particular for wood and wood-based panels [4], on which wood price had an influence as well. Different wood species are used for the production of wood-based panels, depending on their availability. The species of Polish forest consists of ≈70% of pine (*Pinus sylvestris* L.), and ≈6% of oak (*Quercus robur* L.), spruce (*Picea abies* (L.) H. Karst.) and birch (*Betula pendula* Roth) each [5]. For this reason, pine and spruce wood are the most used materials for the production of MDF, while alder (*Alnus glutinosa* (L.) Gaertn.), birch, and beech (*Fagus sylvatica* L.) wood are used rarely [6]. The average price of pinewood, reported by Polish State Forests, has increased by about 15% from $53 m^−3^ in the period 2013–2017 [7]. This situation requires academia and industry within the wood-based products to investigate and develop new sources of raw material or its substitutes. Not only sawdust, but yearly-available straws or plastics (for wood plastic composites–WPC) may be used for board production. Also, recovered wood and recovered PB and MDF boards may be treated and used as a substitute for raw wooden material for the boards production [8].

Particleboards are the most common wood-based products currently used in recycled wooden materials for their production (WRAP, www.nottinghamshire.gov.uk) [9]. Previous work has explored the production of medium density particleboards (MDP) with wood waste from construction and demolition [10]. As an example, Yang (2007) [11] studied the use of recycled wood-waste chips for the production of particleboards for use in kitchens, bathrooms, as a flooring material, and in outdoor equipment. Moreover, Laskowska & Mamiński in 2018 showed that it is possible to substitute 20–100% of virgin material with post-industrial plywood [12]. In another work, the addition of 30–50% of recovered MDF into PB production was proven possible but with adverse influences on the mechanical and physical properties of the final products [13].

Similarly to particleboards, it is possible to reintroduce wood waste fibrous material when producing fiberboards. Previous works have explored the elaboration of insulation fiberboards, as soft boards (SB) [14] using recovered fibers. Another work has elaborated hardboard (HB) made of recycled corrugate cardboard [15], reintroducing 20% of recovered MDF for dry-process MDF production, where panels meet the requirements of relevant standard requirements [16]. Furthermore, to reduce wood wastes disposed of in landfills, MDF can be produced from up to 100% recovered fibers, or even of old newsprint fibers [17,18]. Also, there are suggestions to re-use the post-used MDF fibers in MDF production after their steam refining conversion [19]. This approach sounds to be more reasonable than the deeper processing of the MDF recovered fibers to bioethanol production [20]. However, these MDF end-of-life scenarios also should be applied instead of landfills.

Although the recycling of wooden materials is environmentally friendly [8], the recovered fiber pulp has different chemical and physical characteristics when compared to virgin fibers [21]. These differences affect the final mechanical and physical properties of the fiberboards negatively [20]. Due to that fact, the process requires an optimization of the production parameters to obtain the best features of fiberboards produced with waste panels returned into the production as a raw material. Moreover, to achieve this, the physical and mechanical properties of reversed materials must be known [22]. Fibers produced from waste wood material, such as particleboards, OSB, and MDF, are suitable and may be used as secondary raw material for the production of fiberboards [23]. In this sense, the examination of fibers recovered from fiberboards can be a useful tool for assessing the potential properties of recycled fiberboards [24].

According to data provided by FAO (Food and Agriculture Organization) [2], the production of fiberboards (MDF and HDF) in Poland increased nearly five times over the last 20 years from 768,000 m^3^ in 2000 to over 3,600,000 m^3^ in 2018. One of the most significant impacts on this growth was the opening of ultrathin HDF production lines, such as Kronospan in 2003 [25], Homanit Karlino in 2005, Homanit Krosno Odrzańskie in 2015 [26], Pfleiderer Grajewo in 2007 [27], and Ikea Industry Orla in 2011 [28]. This increase was a response of wood-based panels producers to increasing demand for HDF boards used in honeycomb construction doors as skins or for furniture production in board-on-frame construction (BOF) [29]. As a conclusion drawn from the information above, the HDF production market in Poland is developing very fast. An increasing wood price caused by an increasing need for wooden material from new wood-based panel factories, as well as the availability [30] of suitable recovered material for fiberboard production, results in the need to evaluate the influence of the share of HDF-r on the final properties of industrial HDF.

The goal of this investigation was to determine the influence of the different amount of recovered HDF on the mechanical properties of industrial high-density fiberboard, such as modulus of rupture, modulus of elasticity, internal bond, and surface soundness, but also on physical properties such as thickness swelling, surface water absorption, surface roughness, and formaldehyde content, as well as on surface color.

## 2. Materials and Methods 

### 2.1. Materials

Virgin pine (*Pinus sylvestris* L.) debarked round wood from Polish State Forests (Podlaskie voivodeship, Poland) was used to produce the fiberboards. Virgin fibers were produced on industrial Metso Defibrator EVO56 (Metso, Helsinki, Finland) with a 2.5 m diameter disc with ten knives. Recovered HDF (HDF-r) came from internal HDF production reject, such as offcuts from side trimming saws and leftovers from the process board breaker, these shredded HDF were mixed with virgin chips in the feeding conveyor of the defibrillator. Raw boards used for this investigation as a target were 2.5 mm HDF with an aimed density of 860 kg m^−3^, and compelled with CARB 2 standard of formaldehyde content. 

### 2.2. Elaboration of HDF 

Four different mass shares of recovered HDF were examined: 2%, 5%, 10%, and 20%. Wood chips or a mix of shredded HDF-r and wood chips were treated under constant hydrothermal parameters. Briefly, a preheating pressure of 0.94 MPa at 180 °C for 3.2 min with average defibrating energy consumption of 145 kWh t^−1^, giving an average bulk density of ≈22.15 kg m^−3^. A paraffin emulsion was added to the defibrillator containing 0.5% of dry paraffin calculated on the weight of the oven-dried fibers. Fibers were glued on a blow-line system with high-steam pressure using a commercial melamine-urea-formaldehyde (MUF) resin with a melamine content of 5.2%, molar ratio of 0.89, and solid content of 66.5%. The resination was set at 11% of dry resin calculated on dry fibers with 21% of urea addition and 3.0% of ammonium nitrate hardener, both calculated regarding dry resin content. The final moisture content of the fibrous mat after drying and before the pressing was ≈10.5% +/− 0.3%. All boards were pressed with constant parameters on an industrial Dieffenbacher continuous press system with a 5.3 s mm^−1^ press factor.

### 2.3. Adjustment of HDF-r Dosage

To control the proper amount of HDF-r being added into the production of HDF production, the screw feeder was loaded with HDF offcuts and leftovers. Different settings of maximum rotation per minute were put into the controller: 0%, 0.3%, 1%, 2%, 3%, 5%, and 8%. The amount of fed HDF-r particles was weighted each hour, and based on the moisture content, dry tons per hour were calculated for each setting. Based on those measurements, an average feeding capacity was determined and referred to a constant defibrillator capacity of 30 t at h^−1^ (dry-tons per hour). For further analysis, setting on the screw feeding controller for dosing HDF-r. The elaborated HDF had a content of 0% (V0), 2% (V2), 5% (V5), 10% (V10), and 20% (V20) of HDF-r. 

### 2.4. Raw Material Fraction

The fraction of pine chips and HDF-r particles was examined with an IMAL vibrating laboratory sorter with nine sieves. The selected sieve sizes were 40, 20, 10, 8, 5, 3.15, 1, 0.315, and <0.315 mm. The amount of material for each fraction was ≈100 g, and the set time of continuous vibrating was 5 min. Gathered results were shown as an average of three examinations.

### 2.5. Fibers Fraction

A fraction of fibers produced with different HDF-r share was examined with an ALPINE Air Jet Sieve e200LS (Hosokawa Alpine AG, Augsburg, Germany), according to DIN 66165. Six sieves in the sizes of 125, 315, 630, 1000, 1600, and 2500 µm were used. Five grams of each fiber fraction was analyzed; the sieving time was set at 2 min. Gathered results are shown as an average of three examinations.

### 2.6. HDF Examination

All the elaborated HDF were conditioned at 20 °C, an test specimens were cut accordingly to EN-326-2 [31] and EN-326-1 [32]. The moduli of rupture (MOR) and elasticity (MOE) were determined according to EN 310 [33], internal bond (IB) was determined according to EN 319 [34], and surface soundness (SS) was determined according to EN 311 [35]. All the mechanical properties were examined with an IB700 (IMAL) laboratory-testing machine (Imal s.r.l., San Damaso, Italy). Board density was determined according to EN 323 on the IB700 testing machine (Imal s.r.l., San Damaso, Italy), moisture content according to EN 322, and thickness swelling (TS) due to EN 317 [36] while surface water absorption due to EN 382-1 [37]. The test of surface roughness was performed with a Surtronic 25 equipment (Taylor Hobson, Leicester, England) and results are an average from 10 measurements. The density profiles of the HDF were measured on a GreCon DAX 5000 device (Fagus-GreCon Greten GmbH & Co. KG, Alfeld/Hannover, Germany) [38]. Board formaldehyde content was examined according to EN 12460-5 [39] with a Hach-Lange (HACH LANGE GmbH, Duesseldorf, Germany) spectrophotometer. A Konica Minolta (Konica Minolta Co., Ltd, Tokio, Japan) CM-700d/600d unit was used to define HDF surface color according to the CIELab color system. For each board type, as many as ten samples were analyzed.

### 2.7. Statistical Analysis

Analysis of variance (ANOVA) and t-tests calculations were used to test (α = 0.05) for significant differences between factors and levels, where appropriate, using IBM SPSS statistic base (IBM, SPSS 20, Armonk, NY, USA). A comparison of the means was performed when the ANOVA indicated a significant difference by employing the Duncan test.

## 3. Results

The results of the sieving are shown in Table 1. As it could be noticed, the fraction of pine chips was according to the requirements. The sieving of HDF-r particles was comparable to virgin pine chips, but some differences in sieves 2 and 3 could be noticed. There was about 38% less material on sieve 2, which resulted in not meeting the requirements. On the other hand, there was 24% more material on sieve 3. 

Figure 2 presents a visual assessment of virgin pine chips and shredded HDF particles. The first visible difference is the color, with HDF-r being darker than wood chips. These color differences might be caused by the hydrothermal treatment during the defibrating process, but also by the caramelization of simple sugars (pentoses and hexoses) during pressing operation [6].

Although the parameters of the defibrator were kept constant, the bulk density of the fibers differed, as shown in Table 2. Properties of wood fibers as length, distribution, and bulk density have an important influence [40] and depend on a proper defibration. Results in Table 2 show the bulk density measurement of the different fibrous masses with various content of HDF-r. Fibers produced from virgin pine chips had the lowest bulk density compared to those with different HDF-r share. V0 had 18.75 kg m^−3^, while the addition of 2% of HDF-r caused an increase in the bulk density of ≈11% to 21.15 kg m^−3^. An increase in the HDF-r share to 5% caused additional growth in the bulk density of ≈9% (up to 23.31 kg m^−3^). The bulk density then stabilizes, as the difference between 5 and 10% is minimal. In comparison, the addition of 20% HDF-r slightly increased the bulk density to 24.20 kg m^−3^, which represents an increase of less than 4% when compared with the bulk density of fibers having a 10% HDF-r share.

Fibers were analyzed with a fiber-sieving device before the elaboration of composite panels; the results of mass share [%] of different fractions are shown in Table 3. The sieve retaining the majority of the fibers was 125 µm, the percent of fibers retained in this sieve was comparable for V0, V2, and V5, while the lowest value on this sieve was in V10 (60.3%). The main difference can be appreciated on the second sieve (315 µm) in which the number of fibers on sieve size was decreasing with the addition of HDF-r. While V0 had a share of 27%, V10 had 17.7%, representing 34 percentage points less. On the other hand, the sum of fibers from sieves 630, 1000, 1600, and 2500 µm was increasing as the share of HDF-r increased. The total amount of fibers from V0 on the last four sieves was 9.5%, while for V2, V5, and V10, it was 11.6%, 15.82%, and 21.63%, accordingly. As can be seen, the share of fibers in larger sieves for V10 was more than double compared to V0. This might influence the final properties of the HDF. Since HDF boards produced from fibers with a 20% share of HDF-r had many blisters (steam blows in the middle of the thickness of the panels) during production, they could not be examined. For this reason, they were automatically diverted into the board breaker by an online GreCon UPU blister detection sensor. 

The HDF surface color differences have been determined according to the CIELab color system [41]. The results are shown in Figure 3. For this analysis, V0 was used as the reference. According to Commission Internationale de l’Eclairage (CIELab), an important parameter in comparing and analyzing color is ΔE, as it tells about how much the color of the tested sample differs from the standard [42]. In this sense, all tested boards produced with HDF-r addition had noticeable (ΔE > 5) color difference compared to V0. The main aspect was that the addition of HDF-r made the surface darker. The brightest surface was V0, with an L value of 67.96. The biggest difference could be seen for V2 (ΔE = 8.56), and this was related to a more reddish (Δa = 1.75) and yellow (Δb = 3.92) color compared to V0. The lowest color difference is that of V10 (ΔE = 5.30), although the difference was 7% darker. The surface of the V0 had an increase in red (Δa = 1.47) and blue (Δb = 1.58) spectra. V5 varied from V0 (ΔE = 5.36). However, it was comparable to V10. Moreover, V5 changed minimally to a shade of red (Δa = 1.54) and blue (Δb = 1.37).

The density profile distribution was characterized, and the results are shown in Figure 4. The density of the top side of the board is on the left side of the pictures, and the bottom side surface density is on the right side. The difference between the top and bottom surface density was relatively small, and it was for about 2–3% lower for the bottom side.

All the examined density profiles were similar and characteristic for HDF panels; moreover, from these figures, it can be observed that there was no delamination of the HDF. Figure 5 shows the average maximum surface density (ASD) and the average minimum core density (ACD). The highest ASD was obtained for V2 (1136 kg m^−3^), while the lowest ACD was obtained for V10 and V2, being 840 and 841 kg m^−3^ accordingly. The biggest difference between ASD and ACD was observed for V2. Additionally, similar to the lowest ACD, the minimum ASD was obtained for V10 (1091 kg m^−3^), which is 4% less than the maximum. The difference between ASD and ACD for V0 was 262 kg m^−3,^ while for V5, it was 5% lower and gave the smallest difference in density profile (249 kg m^−3^). The ASD of V5 was comparable to that of V0 and was on the level of 1114 kg m^−3^. It is worth noticing that not only the surface and core densities influence the properties of fiberboards properties, but also the difference between those two parameters [43].

Bending strength results are shown in Figure 6. V0 had a modulus of rupture (MOR) of 55.8 N mm^−2^ with a density of 852 kg m^−3^, which is within EN 622-5 requirements (grey line level on Figure 6) of the average density tolerance of 860 kg m^-3^ +/−7% [44]. Panel properties such as MOE and MOR are positively affected by an increase in density [45]. However, having higher moisture content affects those properties negatively [46]. Although V2 had MOR comparable to V0 (55.8 N mm^−2^), its density (872 kg m^−3^) was 20 kg m^−3^ higher, this might have influenced the final MOR result. In particular, the highest surface density (1136 kg m^−3^) might have affected. Additionally, the moisture content of V2 (5.29%) was more than six percentage points higher than V0 (5.63%), which, according to Ganev et al. [46], might have also influenced the result. 

On the other hand, considering that V2 and V5 have almost the same density (870 kg m^−3^) and similar panel moisture content, the decrease of bending strength of V5 for about 10% is related to the content of HDF-r. However, further increase of HDF-r, as in V10, did not result in further MOR value decrease. When analyzing MOR results statistically (Table 4), it should be pointed out that there are no statistically significant differences between the mean values of the MOR results. All of the tested HDF have met the requirements of EN 622.

Regarding MOE, the highest was that of V2 (4651.4 N mm^−2^), and the lowest was that of V10 (4085.0 N mm^−2^). V2 and V5 had accordingly 12% and 2% (4177.3 N mm^−2^) higher modulus of elasticity values compared to V0 (4097.7 N mm^−2^). These values could be explained by the density of those boards, but on the other hand, within the same range of densities, there is a drop of MOE for V5 compared to V2 of more than 10%. Increasing the amount of HDF-r reduces the fiber coverage with the resin and consequently affects the mechanical properties [18]. Another factor affecting the mechanical properties is bulk density [47]. According to statistical analysis, it should be pointed out that there are no statistically significant differences between the mean values of the MOE results.

Results for the internal bond (IB) of the HDF are presented in Figure 7. As it can be seen, not all the panels met the requirements for HDF stated at EN 622-5 (grey line level on Figure 7a), which demands IB to be above 0.65 N mm^−2^. While V0 had an IB of 1.15 N mm^−2^, the mean IB of panels with HDF-r was significantly lower. The lowest IB was obtained for V10 (0.61 N mm^−2^), and it was nearly 50% lower compared to V0, and ≈25% lower than V5 (0.80 N mm^−2^). Based on fiber bulk density and fiber analysis (Table 2 and Table 3), such fibers may have a smaller particle size compared to HDF fibers from V0. The larger amount of fines increases the surface area of the fibers, which results in decreasing the resin coverage per unit surface area [48]. Hence, the strength of the final panel is lower. V2 obtained 15% lower IB (0.67 N mm^−2^) compared to V5, this might be caused by a difference in the density distribution on the profile. In V2, the average difference between ASD and ACD on the density profile was about 295 kg m^−3^, while in V5, it was: 249 kg m^−3^.

Moreover, V2 had a 3% lower ACD than V5 (864 kg m^−3^). According to Wong, the IB of fiberboards is mostly determined by the core density and is not dependent on processing conditions [43]. Additionally, an increase in the share of HDF-r may decrease mechanical properties because of the way they are fibrillated and have a high fine fiber content [18]. The only statistically significant difference in mean values of IB has been found between V0 and the remaining panel types (Table 4).

Surface soundness (SS) was evaluated, for this, EN standards do not present any requirements, but typical industry requirement for SS is >0.80 N mm^−2^ (grey line level on Figure 7b). Based on the results shown in Figure 7, it can be seen that the highest SS was obtained for V2 (1.51 N mm^−2^), and comparing to V0, which had a SS of 1.09 N mm^−2^, it was more than 27% higher. V5 had a 20% lower surface soundness (1.20 N mm^−2^) compared to V2, but nearly 10% higher than V0. The lowest SS was that of V10 (1.02 N mm^−2^), which was ≈32%, lower than V2. While there is a direct relation between the bulk density of fibers and the mechanical properties of the MDF [49], there is an inversely proportional influence of the HDF-r share on SS. That might be caused by a worse quality of the recovered fibers as a consequence of their decrease in length (above 1711 μm) and also a significant increase in smaller fibers (with length 200–956 μm) compared to virgin pine fibers [24]. There are no statistically confirmed differences between the mean values of the surface soundness of tested panels (Table 4).

The thickness swelling (TS) after 2 and 24 h values of HDF panels are displayed in Figure 8. The results for V0, V2, and V5 showed that TS2 and TS24 were dependent on the share of HDF-r, having a positive linear correlation. Wood, but also all wood-based products, shrink or swell depending on the moisture content of the material [50]. The relationships between shrinkage, swelling, and moisture content are roughly proportional [51], and there have been several reviews in the literature confirming that swelling of wood-based panels is decreasing with an increase of the moisture content [52]. For this reason, the minimum TS2 and TS24 could be noticed for V0, being 9.90% and 31.03%, accordingly, while the moisture content of V0 was 5.63%, which was the highest observed. The TS24 of V2 (32.82%) increased ≈5% compared to V0, while V5 had an increase of ≈5% to 34.25%. An increase in TS of panels with HDF-r might be caused by their rougher structure and their smaller dimensions compared to virgin fibers [21,24]. The highest TS2 and TS24 were measured for V10, being 11.71% and 34.54%, respectively, which was ≈15% and ≈10% higher than V0. As can be seen, all the panels met the requirements for HDF stated at EN 622-5 (grey line level on Figure 8a). The only statistically significant differences between the achieved mean values of thickness swelling, both TS2 and TS24, have been confirmed between V0 and V10 (Table 4).

The surface water absorption (WA) was also examined. Although thickness swelling after 24 h was increasing with HDF-r share, reaching the highest value for V10 and the lowest for V0, the WA had the opposite trend. As can be seen in Figure 8, the highest WA for top and bottom HDF side was obtained for V0, accordingly, 255 g m^−2^ and 273 g m^−2^, while the lowest was for V10, with results of top side WA of 175 g m^−2^ and bottom side WA of 185 g m^−2^. All panels with HDF-r had ≈30% lower top (V2 = 187 g m^−2^; V5 = 186 g m^−2^) and bottom (V2 = 191 g m^−2^ V5 = 187 g m^−2^) surface water absorption compared to V0, which is consistent with other works [21]. Moreover, in all panels, higher WA could be observed for the bottom side compared to the top side. For V0, it was 7% higher, for V10, 5% higher, and V2 and V5, 1.5% higher as an average. 

The surface roughness was examined since one of the factors influencing the consumption of sealing materials during lacquering is the roughness of the HDF surface, increasing together with the increase of the roughness [53]. Based on the results shown in Figure 8, the higher WA of HDF bottom side might be caused by a more closed surface (14%) of the top side for all the panels, which had an average roughness of 3.09 µm, compared to an average roughness of the bottom side of 3.69 µm. This can be connected to a higher density of the top side layer, which was confirmed in the Figure 4. Moreover, the lower surface roughness of panels produced with HDF-r could be caused by a higher fiber density compared to V0, which is easier to compress during the pressing process.

As it was mentioned, the panels were produced in compliance with CARB 2 standard. Therefore, the required formaldehyde content should be below 5.0 mg 100 g^−1^ (grey line level on Figure 9b). In this sense, not all the boards met that requirement being V2 and V5 out of the CARB 2 specifications. As can be seen in Figure 9b, V0 had the lowest formaldehyde content (3.76 mg 100 g^−1^), while the highest was obtained for V2 (5.37 mg 100 g^−1^), which is ≈43% more. Further addition of HDF-r caused a decrease in formaldehyde content in V5 to 5.21 mg 100 g^−1^ but still out of specification. However, V10 had the lowest result of formaldehyde content (4.98 mg 100 g^−1^) from all the tested boards produced with HDF-r, which allowed meeting CARB 2 requirements. This decrease of formaldehyde content with an increase in the recycled share of fibers was also observed for MDF [21].

## 4. Conclusions

Recovered HDF is a suitable raw material substitute as a complement to virgin pine chips for industrial HDF production. It may be possible to produce industrial HDF with up to 10% HDF-r addition and meet EN 622-5 standard selected requirements. The surface of HDF produced with 2% HDF-r is significantly darker compared to those made of virgin pine wood. The addition of HDF-r caused an increase in fiber bulk density, which has an impact on the performance of HDF. Increasing the HDF-r share from 0% to 10% caused a decrease of MOR for about 9% while adding 2% of HDF-r caused an increase of MOE of ≈12%, and further addition of HDF-r caused a slight decline of MOE. Increasing HDF-r share from 0% to 10% caused a decrease of internal bond for about 47%, though these boards did not meet EN standard requirements. Implementation of 2% of HDF-r resulted in higher surface soundness for about 28%, while a further increase of HDF-r to 10% caused a decrease of MOE of ≈6%. The addition of HDF-r represented a 30% lower surface water absorption and about 12% lower surface roughness. An increase in the share of HDF-r represented an increase in the thickness swelling after 24 h. The addition of HDF-r represented an increase of formaldehyde content, which might imply a reformulation of the bonding agent to meet with the CARB 2 standard.

## Figures and Tables

**Figure 1 polymers-12-01327-f001:**
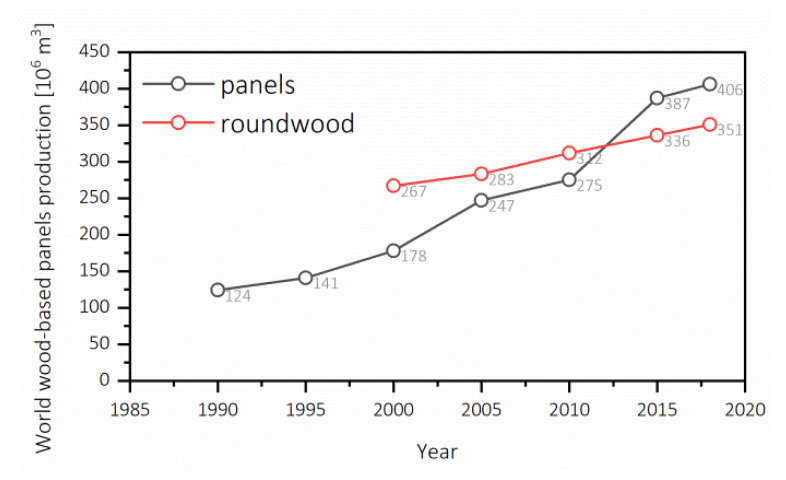
World wood-based panels production 1990-2018 and World pulpwood, round and split, coniferous production 2000–2018 (own elaboration according to [2]).

**Figure 2 polymers-12-01327-f002:**
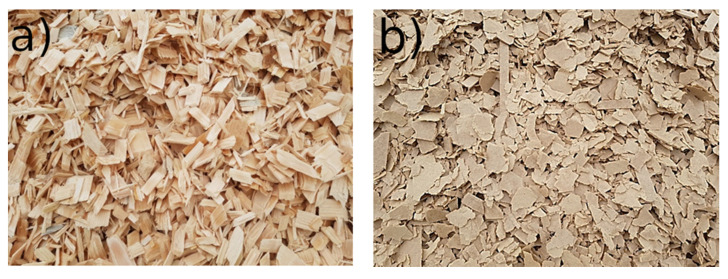
Pictures of produced chips: virgin pine (**a**) and HDF-r (**b**).

**Figure 3 polymers-12-01327-f003:**
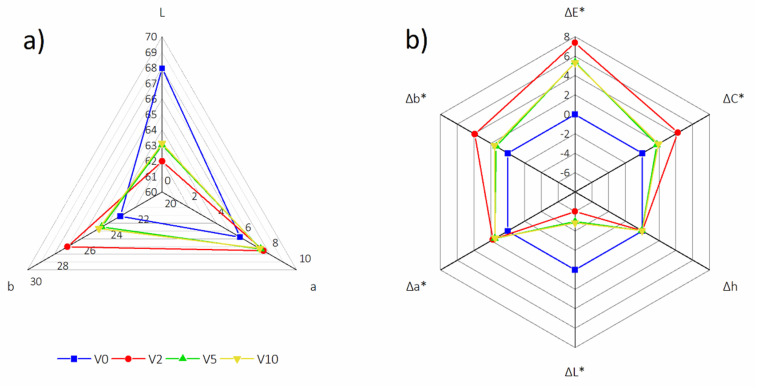
HDF color analysis: L, a and b parameters (**a**) and E, C, h, L, a and b changes (Δ) (**b**).

**Figure 4 polymers-12-01327-f004:**
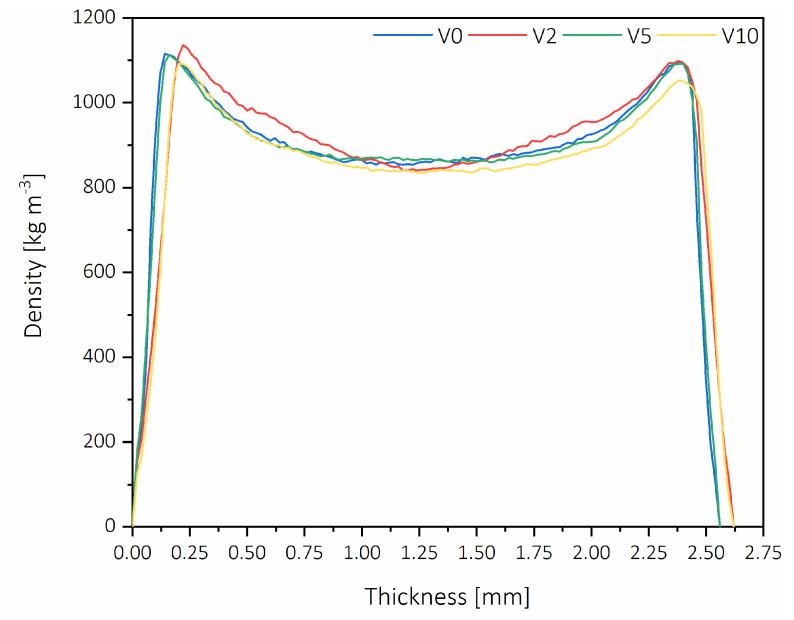
Vertical density profiles of tested HDF panels.

**Figure 5 polymers-12-01327-f005:**
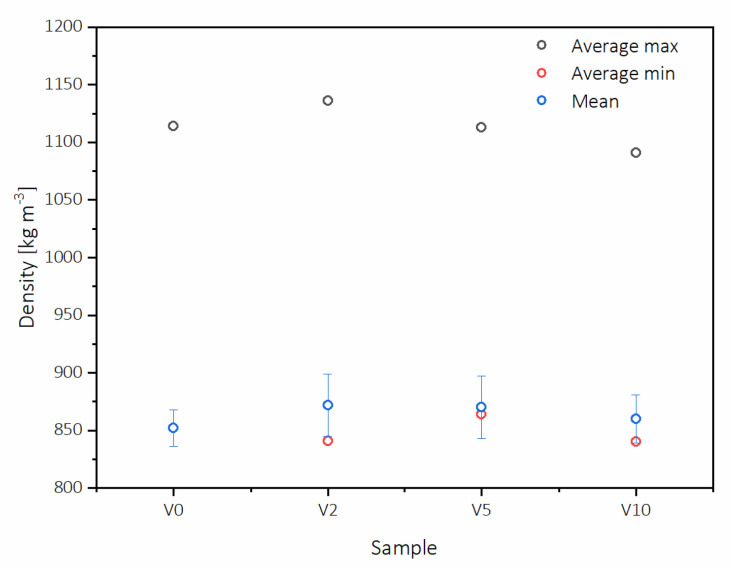
HDF boards average maximum surface density and average minimum core density.

**Figure 6 polymers-12-01327-f006:**
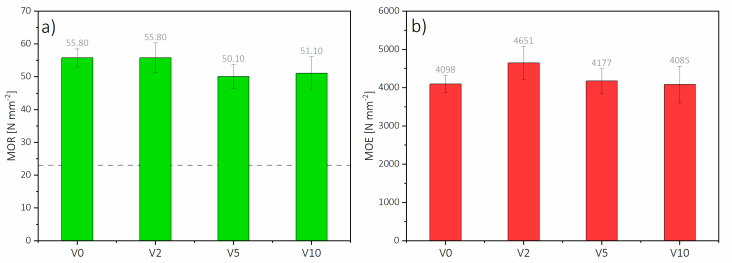
(**a**) HDF Modulus of rupture (MOR), and (**b**) Modulus of Elasticity (MOE).

**Figure 7 polymers-12-01327-f007:**
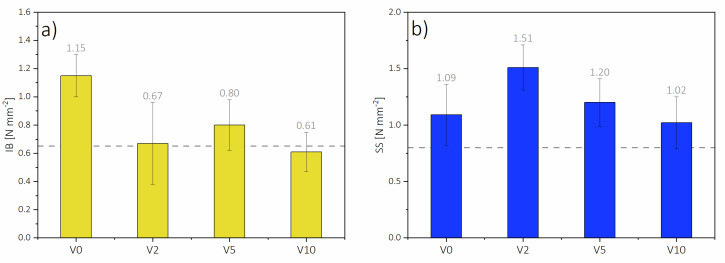
(**a**) Internal bond (IB) and (**b**) surface soundness (SS) results.

**Figure 8 polymers-12-01327-f008:**
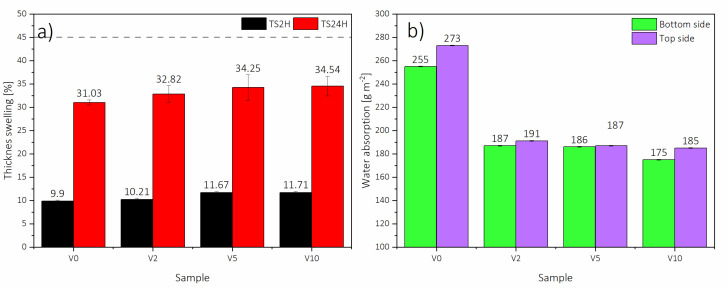
(**a**) HDF Thickness Swelling after 2h and 24h, and (**b**) Surface Water Absorption (WA) in the top and bottom sides.

**Figure 9 polymers-12-01327-f009:**
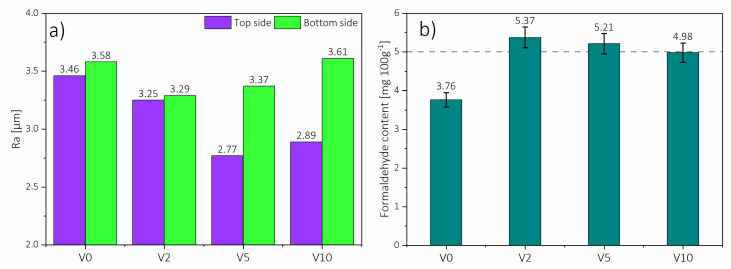
HDF surface roughness of top and bottom sides (**a**) and formaldehyde content of the elaborated HDF (**b**).

**Table 1 polymers-12-01327-t001:** Sieving results for pine and spruce wood chips.

Sieve	1 ^a^	2 ^b^	3 ^c^	4 ^d^	5 ^e^	6 ^e^	7 ^f^	8 ^f^	9 ^f^
Size [mm]	40	20	10	8	5	3.15	1	0.315	<0.315
Pine chips [%]	0.74	51.80	39.19	4.64	3.07		0.56		
Recycled HDF [%]	0.55	32.48	52.50	4.67	4.15		0.50		

Requirements: ^a^ <1%, ^b^ 45%<, ^c^ 35%<, ^d^ <5%, ^e^ <8%, ^f^ <1%.

**Table 2 polymers-12-01327-t002:** Fibers bulk density for panels.

Sample	V0	V2	V5	V10	V20
**Fiber bulk density [kg m^−3^]**	18.75	21.15	23.31	23.33	24.20

**Table 3 polymers-12-01327-t003:** Fractions share of tested fibers mass.

Sieve Size [µm]	Variant			
	V0	V2	V5	V10
**125**	63.5	63.6	62.2	60.3
**315**	27.0	24.8	21.8	17.7
**630**	7.9	6.6	8.6	9.3
**1000**	1.4	3.6	4.8	9.2
**1600**	0.2	1.3	2.4	3.1
**2500**	0.0	0.1	0.2	0.4

**Table 4 polymers-12-01327-t004:** Statistical analyses results (*p*-values).

**MOR**	**V2**	**V5**	**V10**	**MOE**	**V2**	**V5**	**V10**
V0	0.287	0.067	0.078	V0	0.052	0.115	0.287
V2		0.286	0.081	V2		0.064	0.073
V5			0.242	V5			0.279
**IB**	**V2**	**V5**	**V10**	**SS**	**V2**	**V5**	**V10**
V0	***0.004 ****	***0.046***	***0.003***	V0	0.056	0.119	0.268
V2		0.282	0.276	V2		0.054	0.051
V5			0.084	V5			0.243
**TS 2 h**	**V2**	**V5**	**V10**	**TS 24 h**	**V2**	**V5**	**V10**
V0	0.254	0.058	***0.046***	V0	0.057	0.061	***0.029***
V2		0.069	0.064	V2		0.124	0.138
V5			0.199	V5			0.035

* enhanced values indicate statistically significant differences.

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
