# Peer review of "Influence of Adding Offcuts and Trims with a Recycling Approach on the Properties of High-Density Fibrous Composites"

_polymers, 2020, doi:10.3390/polym12061327_

Round 1

Reviewer 1 Report

please see my comments in the attached file

Reviewer 2 Report

Line 54. While comparing costs in monetary terms, this information tends to age bad. I suggest transforming the information to percentage increase of the costs.

Line 109 and 117. The authors use a “defibrillator” with the wood chips. I must admit that the mental image was fun, but I suggest to use a “defibrator”.

Table 1. In my opinion the information is unclear. I suggest. Drawing an horizontal line below size, as it is not a result. Add [%] after Pine chips and Recycle HDF, to instantly know that the results are percentages.

Figure 2, line 174. The change in color is clear, but at first sight, recycled chips seem to be more planar than virgin wood.

Line 183. I think that table 3 shows the results, but not a correlation, understood as statistical correlation. Please change that word to avoid misunderstandings.

There are two tables with number 2.

Second table 2. Line 194. The authors say that the values are presented as “w/w”, with a maximum value of 1. I think that the results are presents as “% w/w”.

Line 222. The authors present figure 4 and then in the next sentence figure 5. If the authors can only present the figure without further comments. This figure can be deleted. Please, comment figure 4 to add value or delete it. In its present shape I understand that after introducing figure 5, all the comments are related with this figure. Otherwise, change the text or present only a figure 4 with 4a and 4b.

Figure 6. Mark 6a and 6b. Idem figure 7

Figure 7. I suggest drawing an horizontal line signaling the 0.65 N EN 622-5 limit. The same can be made for the industry requirement SS 0.80 N mm-2.

A final recommendation. The authors present a lot of figures, this is in my opinion good, because graphic information is easy to read and rich. Nonetheless, the authors use a lot of different styles (marker shapes) and colors. I suggest to homogenize the colors and the shapes to present a more elegant paper. But this is my own opinion based on my contact with graphic designers.

Round 2

Reviewer 1 Report

The authors addressed all my comments and I am happy to suggest the paper to be accepted in its current form